# JARVIS: A Neuro-Symbolic Commonsense Reasoning Framework for Conversational Embodied Agents

**Kaizhi Zheng**[*]                                    KZHENG31@UCSC.EDU
**Kaiwen Zhou**[*]                                    KZHOU35@UCSC.EDU
**Jing Gu**[*]                                         JGU110@UCSC.EDU
**Yue Fan**[*]                                         YFAN71@UCSC.EDU
**Jialu Wang**[*]                                      JWANG470@UCSC.EDU
**Zonglin Di**                                        ZDI@UCSC.EDU
**Xuehai He**                                         XHE89@UCSC.EDU
**Xin Eric Wang**                                     XWANG366@UCSC.EDU
*University of California, Santa Cruz*

**Editors:** Leilani H. Gilpin, Eleonora Giunchiglia, Pascal Hitzler, and Emile van Krieken

## Abstract

Building a conversational embodied agent to execute real-life tasks has been a long-standing yet quite challenging research goal, as it requires effective human-agent communication, multi-modal understanding, long-range sequential decision making, etc. Traditional symbolic methods have scaling and generalization issues, while end-to-end deep learning models suffer from data scarcity and high task complexity, and are often hard to explain. To benefit from both worlds, we propose JARVIS, a neuro-symbolic commonsense reasoning framework for modular, generalizable, and interpretable conversational embodied agents. First, it acquires symbolic representations by prompting large language models (LLMs) for language understanding and sub-goal planning, and by constructing semantic maps from visual observations. Then the symbolic module reasons for sub-goal planning and action generation based on task- and action-level common sense. Extensive experiments on the TEACh dataset validate the efficacy and efficiency of our JARVIS framework, which achieves state-of-the-art (SOTA) results on all three dialog-based embodied tasks, including Execution from Dialog History (EDH), Trajectory from Dialog (TfD), and Two-Agent Task Completion (TATC) (e.g., our method boosts the unseen Success Rate on EDH from 6.1% to 15.8%). Moreover, we systematically analyze the essential factors that affect the task performance and also demonstrate the superiority of our method in few-shot settings. Our JARVIS model ranks first in the Alexa Prize SimBot Public Benchmark Challenge[1].

## 1. Introduction

A long-term goal of embodied AI research is to build an intelligent agent capable of communicating with humans in natural language, perceiving the environment, and completing real-life tasks. Such an agent can autonomously execute tasks such as household chores, or follow a human commander to work in dangerous environments. Figure 1 demonstrates an example of dialog-based embodied tasks: the agent communicates with the human commander and completes a complicated task "making a sandwich", which requires reasoning about dialog and visual environment, and procedural planning of a series of sub-goals.

---

[*] Equal contribution

1. https://eval.ai/web/challenges/challenge-page/1450/leaderboard/3644

Although end-to-end deep learning models have extensively shown their effectiveness in various tasks such as image recognition (Dosovitskiy et al., 2021; He et al., 2017) and natural language understanding and generation (Lewis et al., 2020; Dathathri et al., 2020), they achieved little success on dialog-based embodied navigation and task completion with high task complexity and scarce training data due to an enormous action space (Padmakumar et al., 2021). In particular, they often fail to reason about the entailed logistics when connecting natural language guidance with visual observations, and plan efficiently in the huge action space, leading to ill-advised behaviors under unseen environments. Conventionally, symbolic systems equipped with commonsense knowledge are more conducive to emulating humanlike decision-makings that are more credible and interpretable. Both connectionism and symbolism have their advantages, and connecting both worlds would cultivate the development of conversational embodied agents.

To this end, we propose JARVIS, a neuro-symbolic commonsense reasoning framework towards modular, generalizable, and interpretable embodied agents that can execute dialog-based embodied tasks such as household chores. First, to understand free-form dialogs, a large language model (LLM) is applied to extract task-relevant information from human guidance and produce actionable sub-goals for completing the task (e.g., the sub-goals 1.1, 1.2,...,3.6 in Figure 1). During the process, a semantic world representation of the house environment and object states is actively built from raw visual observations as the agent walks in the house. Given the initial sub-goal sequence and the semantic world representation being built, we design a symbolic reasoning module to generate executable actions based on task-level and action-level common sense.

We evaluate our JARVIS framework on three different levels of dialog-based embodied task execution on the TEACh dataset (Padmakumar et al., 2021), including Execution from Dialogue History (EDH), Trajectory from Dialogue (TfD), and Two-Agent Task Completion (TATC). Our framework achieves state-of-the-art (SOTA) results across all three settings. In a more realistic few-shot setting where available expert demonstrations are limited for training, we show our framework can learn and adapt well to unseen environments. Meanwhile, we also systematically analyze the modular structure of JARVIS in a variety of comparative studies. Our contributions are as follows:

- We propose a neuro-symbolic commonsense reasoning framework blending the two worlds of connectionism and symbolism for conversational embodied agents. The neural modules convert dialog and visual observations into symbolic information, and the symbolic modules integrate sub-goals and semantic maps into actions based on task-level and action-level common sense.

- Our framework is modular and can be adapted to different levels of conversational embodied tasks, including EDH, TfD, and TATC. It consistently achieves the SOTA performance across all three tasks, with great generalization ability in new environments.

- We systematically study the essential factors that affect the task performance, and demonstrate that our framework can be generalized to few-shot learning scenarios when available training instances are limited.

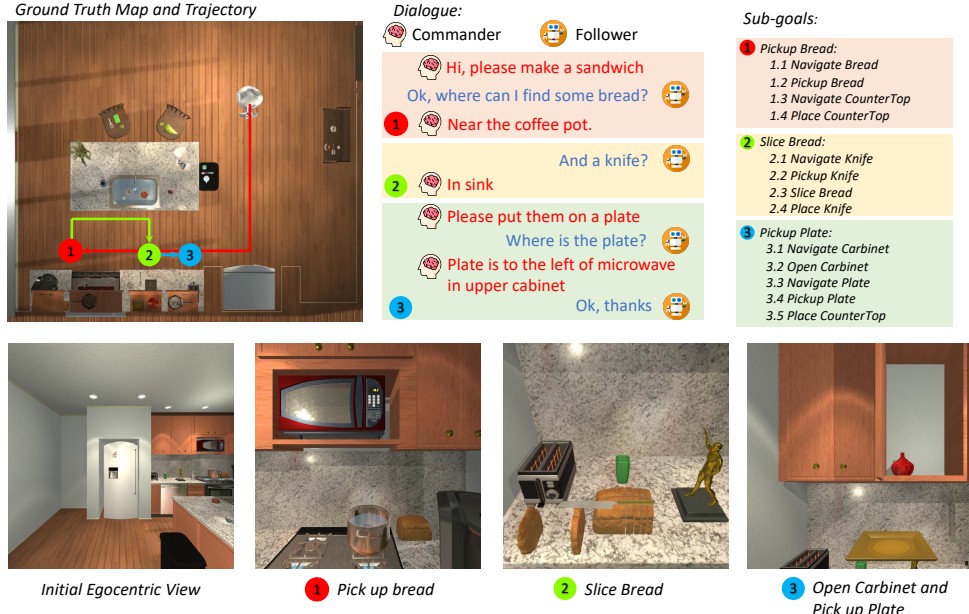

Figure 1: **Dialogue-based embodied navigation and task completion.** The Commander (often a human) issues a task such as making a sandwich, and the Follower agent completes the task while communicating with the Commander. Unlike the agent in fine-grained instruction following tasks, the Follower agent needs to extract sub-goals from the free-form dialogue and execute actions in the visual environment. Note that the Follower agent can only navigate and interact with objects in an egocentric view and has no access to the map or other oracle information.

## 2. Related Work

**Embodied AI Tasks** Embodied agents capable of navigation and interaction have been long studied in AI, with early work focusing on goal-directed navigation in indoor and outdoor environments (Zhu et al., 2017; Yang et al., 2019; Kim et al., 2006). More recently, research has shifted toward language-grounded agents. Vision-and-Language Navigation (Anderson et al., 2018; Ku et al., 2020; Zhu et al., 2020a; Chen et al., 2019; Qi et al., 2020; Vasudevan et al., 2021; He et al., 2021; Gu et al., 2022) explores how agents follow natural language instructions to reach target locations, while Vision-and-Dialog Navigation (Thomason et al., 2019; Banerjee et al., 2020; Nguyen et al., 2019; Nguyen and Daumé III, 2019) incorporates real-time dialogue for guidance. Beyond navigation, recent efforts (Shridhar et al., 2020; Misra et al., 2018; Gordon et al., 2018) introduce interactive task completion involving both navigation and object manipulation. Dialog-based embodied tasks (Padmakumar et al., 2021; Narayan-Chen et al., 2019) further align with real-world settings, enabling agents to collaborate with humans through conversation. Our work builds on this line, aiming to develop a conversational agent for complex household tasks.

**Vision-and-Language Navigation and Task Completion** Prior work in embodied AI has explored vision-and-language navigation (Wang et al., 2019; Hong et al., 2021; Tan et al., 2019; Fried et al., 2018; Guhur et al., 2021; Chen et al., 2021a; Gu et al., 2022), vision-and-dialog navigation (Wang et al., 2020; Zhu et al., 2020b; Kim et al., 2021), and task-oriented

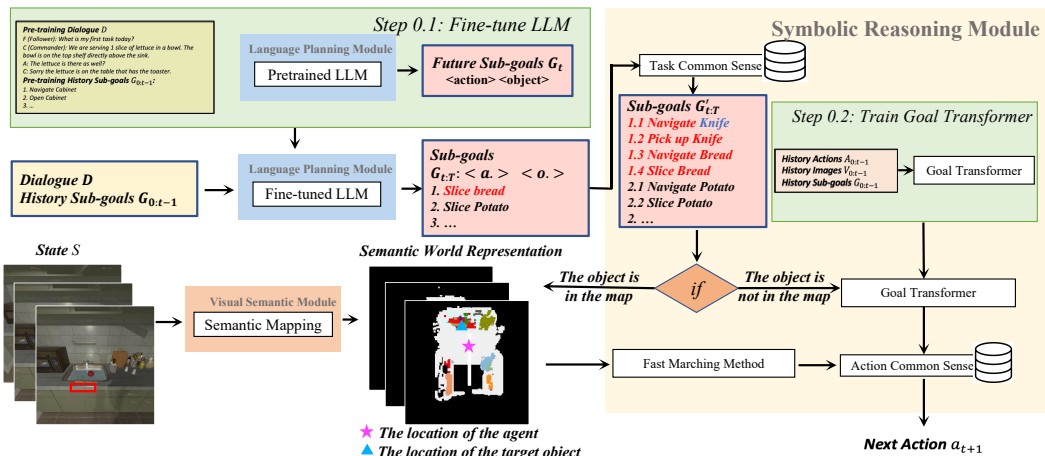

Figure 2: **An overview of our JARVIS framework.** The fine-tuned language planning model takes dialogue and previous sub-goals $G_{0:t-1}$ as input and produces the future sub-goals $G_{t:T}$ (Section 3). $G_{t:T}$ will be further examined by our Task-Level Common Sense model and converted to more reasonable and detailed future sub-goals $G'_{t:T}$. Meanwhile, the Visual Semantic module actively updates the semantic world representations (Section 3). If the object is found in the world representation, the next action is determined by the Fast Marching method. If not, the Goal Transformer will generate the next action. The next action $a_{t+1} \in \mathcal{A}$ will be post-processed by the Action-Level Common Sense model.

object navigation (Ramrakhya et al., 2022). Vision-and-language task completion has also gained attention (Pashevich et al., 2021; Min et al., 2022; Blukis et al., 2021; Song et al., 2022), with early methods like Shridhar et al. (2020) using LSTMs for multi-modal fusion, and Pashevich et al. (2021) introducing transformers for improved temporal modeling. Modular approaches further decompose tasks: Blukis et al. (2021) use semantic voxel grids with hierarchical controllers, and Min et al. (2022) integrate semantic policies for fine-grained object search. However, these systems are designed for structured, instruction-following tasks and often struggle with the ambiguity and complexity of dialog-based scenarios. In contrast, our approach extracts task-relevant information directly from free-form dialogue and performs neuro-symbolic reasoning to generalize across diverse dialog-based embodied tasks.

**Neuro-Symbolic Reasoning** While neural models have achieved great success across vision and language tasks (He et al., 2017; Devlin et al., 2019; He et al., 2016; Dosovitskiy et al., 2021), they often lack interpretability and reasoning ability in complex, multi-modal settings. Neuro-symbolic methods address this by combining neural perception with symbolic reasoning (Mao et al., 2019; Chen et al., 2021b; Gupta et al., 2020; Hudson and Manning, 2019; Sakaguchi et al., 2021). For example, Mao et al. (2019) introduce a symbolic program generator for visual question answering, and Chen et al. (2021b) extend it to video reasoning. In dialog-based household tasks, neural models struggle to integrate diverse inputs and infer correct actions (Padmakumar et al., 2021). To address this, we propose a modular neuro-symbolic framework that converts multi-modal observations into symbolic representations and applies commonsense reasoning for robust task execution.

## 3. Neuro-Symbolic Conversational Embodied Agents

### Problem Formulation

We study dialogue-based embodied agents, where a *Follower* agent must interpret natural language instructions from a *Commander* (a human or another agent) to complete long-horizon tasks in a visual environment. Each task begins from an initial state $s_i \in \mathcal{S}$ and proceeds through multi-turn dialogue $D = \{(p_i, u_i)\}$, where $p_i \in \{\text{Commander}, \text{Follower}\}$ indicates the speaker and $u_i$ is the utterance. At each timestep $t$, the *Follower* receives a visual observation $v_t$ and executes an action $a_t \in \mathcal{A} : \mathcal{S}_t \to \mathcal{S}_{t+1}$, aiming to reach a goal state $s_f \in \mathcal{S}$.

Tasks often involve intermediate sub-goals (e.g., "Find bread", "Cook egg") under a high-level goal (e.g., "Make breakfast"). The *Commander* has oracle access to task and environment details, while the *Follower* has only egocentric perception and can request clarification during the dialogue.

Following Padmakumar et al. (2021), we consider three settings: in Execution from Dialog History (EDH), the agent completes an unfinished task given past dialogue and partial trajectory; in Trajectory from Dialog (TfD), the agent reconstructs a full action sequence from complete dialogue; and in Two-Agent Task Completion (TATC), the agent collaborates interactively with a *Commander* to complete the task. Our proposed JARVIS framework is designed to handle all three scenarios.

### Proposed Methods

Our JARVIS framework (as shown in Figure 2) consists of a Language Planning module, a Visual Semantic module, and a Symbolic Reasoning module. Specifically, the Language Planning module utilizes a pre-trained large language model to process free-form language input and produces procedural sub-goals of the task. In the Visual Semantic module, we use a semantic segmentation model, a depth prediction model, and the SLAM algorithm to transform raw visual observations into a more logistic form — semantic maps containing spatial relationships and states of the objects. Finally, in the Symbolic Reasoning module, we utilize task-level and action-level commonsense knowledge to reason about transferred symbolic vision-and-language information and generate actions, and a Goal Transformer is trained to deal with uncertainty by directly producing actions when no relevant information can be retrieved from the visual symbolic representations. Below we introduce our methods in detail; please refer to Appendix A for more implementation details and Appendix B for our notation table.

**Language Understanding and Planning** The language commands in dialog-based embodied tasks are free-form and high-level, and do not contain low-level procedural instructions. Therefore, it is essential to break a command like "can you make breakfast for me?" into sub-goals such as "Find bread", "Find knife", "Slice bread", "Cook egg", and then generate actions $a_i$ to complete the sub-goals sequentially. Large language models (LLMs) are shown to be capable of extracting actionable knowledge from learned world knowledge (Huang et al., 2022). In order to understand free-form instructions and generate the sub-goal sequence for action planning, we leverage an LLM, the pre-trained BART (Lewis et al., 2020) model, to process dialog and action, and predict future sub-goal sequence

$G_{t:T}$ for completing the whole task:

$$\widehat{G}_{t:T} = \text{LLM}(D, G_{0:t-1}) \tag{1}$$

where $D = \{(p_i, u_i)\}$ is the collected set of user-utterance pairs and the previous sub-goal sequence $G_{0:t-1} = \{g_0, g_1, g_2, ..., g_{t-1}\}$. For sequential inputs $G_{0:t-1}$, we encode the sub-goals into tokens and concatenate them as the input of the LLM model. For the ground-truth sub-goal sequence $G_{0:T}$, we acquire it by a rule-based transformation from the action sequence $A_{0:T} = \{a_0, a_1, a_2, ..., a_t, ..., a_T\}$. Concretely, we note that the actions can be categorized as navigations and interactions. For interactions, we coalesce the action and targeted object as the sub-goal. For example, if the embodied agent executes the action of picking up a cup, we will record the sub-goal "PickUp Cup". For navigation, we coalesce "Navigate" with the target object of the next interaction.

Note that in cases like the Trajectory from History (TfD) task, where only dialog information $D$ is given and other history information is missing, Equation 1 reduces to

$$\widehat{G}_{t:T} = \text{LLM}(D) \tag{2}$$

The BART model is trained with a reconstruction loss by computing the cross entropy between the generated future sub-goal sequence $\widehat{G}_{t:T}$ and the ground truth future sub-goal sequence $G_{t:T}$.

**Semantic World Representation**   The visual input into a embodied agent is usually a series of RGB images $V = \{v_0, v_1, v_2, ..., v_T\}$. One conventional way in deep learning is to fuse the visual information with other modalities into a neural network (e.g., Transformers) for decision making, which, however, is uninterpretable and often suffers from poor generalization under data scarcity or high task complexity. Thus, we choose to transform visual input into a semantic representation similar to Blukis et al. (2021) and Min et al. (2022), which can be used for generalized symbolic reasoning later.

At each step $t$, we first use a pre-trained Mask-RCNN model (He et al., 2017) for semantic segmentation, which we fine-tune on in-domain training data of the TEACh dataset, to get object types $O_t = \{o_0, o_1, o_2, ..., o_k\}$ and semantic mask $M_t = \{m_0, m_1, m_2, ..., m_k\}$ from egocentric RGB input $v_t$ at time stamp $t$. Then we use a Unet (Ronneberger et al., 2015) based depth prediction model as Blukis et al. (2021) to predict the depth of each pixel of the current egocentric image frame. Then, by combining the agent location and camera parameters, we transform the visual information into symbolic information: if a certain object exists in a 3D space, which we store in a 3D voxel. Then we further project the 3D voxel along the height dimension into a 2D map and obtain more concise 2D symbolic information. So far, we have transformed the egocentric RGB image into a series of 2D semantic maps, among which each map records a certain object's spatial location in the projected 2D plane. This symbolic environment information will be maintained and updated during the task completion process as in Figure. 2.

**Action Execution via Symbolic Commonsense Reasoning**   Once the sub-goal sequence and semantic world representation are obtained, the agent must generate executable actions. However, these inputs may be noisy—sub-goals can be misordered or infeasible,

Table 1: Results in percentages on the EDH and TfD validation sets, where trajectory length weighted (TLW) metrics are included in [ brackets ]. For all metrics, higher is better.

| | EDH | | | | TfD | | | |
|---|---|---|---|---|---|---|---|---|
| | Seen | | Unseen | | Seen | | Unseen | |
| Model | SR [TLW] | GC [TLW] | SR [TLW] | GC [TLW] | SR [TLW] | GC [TLW] | SR [TLW] | GC [TLW] |
| E.T. (Padmakumar et al., 2021) | 8.4 [0.8] | 14.9 [3.0] | 6.1 [0.9] | 6.4 [1.1] | 1.0 [0.2] | 1.4 [4.8] | 0.5 [0.1] | 0.4 [0.6] |
| **Ours** | **15.1 [3.3]** | **22.6 [8.7]** | **15.8 [2.6]** | **16.6 [8.2]** | **1.7 [0.2]** | **5.4** [4.5] | **1.8 [0.3]** | **3.1 [1.6]** |
| E.T. (few-shot)[2] | 6.1 [1.0] | 4.7 [2.8] | 6.0 [0.9] | 4.8 [3.6] | 0.0 [0.0] | 0.0 [0.0] | 0.0 [0.0] | 0.0 [0.0] |
| **Ours (few-shot)** | **10.7 [1.5]** | **15.3 [5.3]** | **13.7 [1.7]** | **12.7 [5.4]** | **0.6 [0.0]** | **3.6 [3.0]** | **0.3 [0.0]** | **0.6 [0.1]** |

and the semantic map may be incomplete. To address this, we introduce a Symbolic Reasoning module that leverages task-level and action-level commonsense logic to validate and refine execution plans. Sample logic predicates are listed in Table 5 (Appendix A).

Task-level reasoning enforces logical preconditions between sub-goals. For instance, the predicate $Pick(agent, x)$ holds only if $x$ is movable and the agent is free-handed. Sub-goals violating such constraints are revised—e.g., inserting "PickUp knife" before "Slice bread." This process assumes ideal execution and updates the agent's internal state accordingly.

Given validated sub-goals and the semantic map, we employ two action generation methods. If the target is visible, the Fast Marching Method (FMM) (Sethian, 1999) plans a path to the nearest empty space near the object. Otherwise, we use a Goal Transformer (GT), adapted from the Episodic Transformer (Pashevich et al., 2021), trained on TEACh data. GT predicts the next action from past observations $V_{0:t-1}$, actions $A_{0:t-1}$, and sub-goals $G_{0:t-1}$:

$$\hat{a}_t = \text{GT}([V_{0:t-1}, A_{0:t-1}, G_{0:t-1}]). \tag{3}$$

Action-level reasoning ensures that planned actions respect environmental constraints. For example, $Move(x, y)$ is valid only if a path exists from $y$ to $x$. If constraints are violated, the agent adapts—e.g., executing a fallback policy like random exploration or navigating to the nearest feasible location.

## 4. Experiments

### Dataset and Tasks

We evaluate JARVIS on the TEACh dataset (Padmakumar et al., 2021), which contains over 3,000 human-human interaction sessions involving household tasks. Each session begins from an initial state $s_i$, proceeds through a multi-turn dialogue $D$ between a *Commander* and a *Follower*, and ends in a final state $s_f$ after executing a reference action sequence $A = \{a_0, a_1, \dots\}$. These sessions form the basis for three evaluation settings:

**Execution from Dialog History (EDH)** requires the agent to complete part of a task by predicting future actions $A_{t:T}$, given the current state $s_i$, dialogue history $D$, and previous actions $A_{0:t-1}$. Success is determined by whether the final simulated state $\hat{s}$ matches the reference $s_f$.

Table 2: Success Rate results on the TATC task with different assumptions of the Commander agent. Our JARVIS establishes the best performance among the current implemented methods.

| | w/ full info. | | w/o GT seg. | | w/o GT & goal loc. | |
|---|---|---|---|---|---|---|
| Model | *Seen* | *Unseen* | *Seen* | *Unseen* | *Seen* | *Unseen* |
| E.T.[3] | 0.0 [0.0] | 0.0 [0.0] | 0.0 [0.0] | 0.0 [0.0] | 0.0 [0.0] | 0.0 [0.0] |
| **Ours** | **22.1 [5.8]** | **16.4 [4.2]** | **9.4 [2.1]** | **5.6 [1.9]** | **3.9 [0.5]** | **1.2 [0.1]** |

Table 3: Success Rates of 12 different task categories in the validation set. For EDH, the results are based on relatively shorter EDH instances, while the results for TdD and TATC tasks are from instances with full trajectories.

| | Plant | Coffee | Clean All | X | Y | Boil | Toast N | Slices X | One Y | Cooked | Sndwch | Salad | Bfast |
|---|---|---|---|---|---|---|---|---|---|---|---|---|---|
| EDH | 21.0 | 21.3 | 14.5 | 15.2 | 15.5 | 12.8 | 22.1 | 19.6 | 15.8 | 15.5 | 10.3 | 14.0 |
| TfD | 13.5 | 6.7 | 6.3 | 4.2 | 1.9 | 0.0 | 0.0 | 0.0 | 0.0 | 0.0 | 0.0 | 0.0 |
| TATC | 70.1 | 29.3 | 40.0 | 18.1 | 5.6 | 4.9 | 0.0 | 25.0 | 0.0 | 0.0 | 0.0 | 0.0 |

**Trajectory from Dialog (TfD)** tasks the agent with reconstructing the entire action sequence $A_{0:T}$ from the complete dialogue $D$ and initial state $s_i$, aiming to reach the target final state $s_f$.

**Two-Agent Task Completion (TATC)** models both *Commander* and *Follower* roles. Given an initial instruction, the *Follower* communicates with the *Commander* to complete the task. The *Commander* has oracle access to environment metadata via three APIs: *ProgressCheck*, which lists task-relevant state differences; *SelectOid*, for querying object identifiers; and *SearchObject*, for locating objects. Following TEACh conventions, we implement JARVIS as two interacting agents: the *Commander* uses a Task-Level Commonsense Module to issue instructions, while the *Follower* executes actions and queries for help via an Action Execution Module.

**Experimental Setup**

**Evaluation Metrics** We adopt Success Rate (SR), Goal-Condition Success (GC), and Trajectory Weighted Metrics (TLW) as evaluation metrics. Task success is a binary value, defined as 1 when all the expected state changes $s_f$ are presented in $\hat{s}$ otherwise 0. SR is the ratio of success cases among all the instances. GC Success is a fraction of expected state changes in $s_f$ present in $\hat{s}$ which is in $(0, 1)$ and the GC success of the dataset is the average of all the trajectories. Trajectory weighted SR (TLW-SR) and GC (TLW-GC) are calculated based on a reference trajectory $A_R$ and a predicted action sequence $\hat{A}$. The

---

2. Our reimplementation version of Padmakumar et al. (2021).

3. From the official TATC Challenge: https://github.com/GLAMOR-USC/teach_tatc. At the time of submission, the E.T. baseline fails on all the TATC instances.

Table 4: Ablation studies on the EDH and TfD validation sets. **Language**: the agent directly teleports to the target goals generated by the language language planning module, thus trajectory length weighted metrics do not make sense here. **Executor**: the agent will use the ground truth sub-goals. **Reasoning**: the agent use both ground truth sub-goals and visual information. **No Goal Transformer**: the agent explore random place when it did not find the target object.

| | EDH | | | | TfD | | | |
|---|---|---|---|---|---|---|---|---|
| | *Seen* | | *Unseen* | | *Seen* | | *Unseen* | |
| Model | SR [TLW] | GC [TLW] | SR [TLW] | GC [TLW] | SR [TLW] | GC [TLW] | SR [TLW] | GC [TLW] |
| JARVIS | 15.1 [ 3.3 ] | 22.6 [ 8.7 ] | 15.8 [ 2.6 ] | 16.6 [ 8.2 ] | 1.7 [ 0.2 ] | 5.4 [ 4.5 ] | 1.8 [ 0.3 ] | 3.1 [ 1.6 ] |
| Language (w/ gt executor) | 19.9 [ — ] | 32.9 [ — ] | 18.8 [ — ] | 36.3 [ — ] | 8.4 [ — ] | 24.7 [ — ] | 10.2 [ — ] | 21.5 [ — ] |
| Executor (w/ gt subgoals) | 38.7 [12.5] | 34.7 [17.9] | 30.5 [ 8.5 ] | 27.3 [12.1] | 5.1 [ 1.3 ] | 8.0 [ 4.3 ] | 1.8 [ 0.2 ] | 4.0 [ 0.9 ] |
| Reasoning (w/ gt subgoals & visual) | 62.7 [27.9] | 67.9 [36.7] | 60.9 [26.2] | 63.6 [35.4] | 13.8 [ 4.0 ] | 24.5 [15.5] | 12.6 [ 4.2 ] | 20.7 [14.6] |
| No Goal Transformer | 17.3 [ 3.2 ] | 21.7 [ 9.0 ] | 15.7 [ 1.6 ] | 16.9 [ 6.3 ] | 0.6 [ 0.0 ] | 3.1 [ 1.1 ] | 1.8 [ 0.2 ] | 1.4 [ 0.6 ] |

trajectory length weighted metrics for metric value $m$ can be calculated as

$$\text{TLW-}m = \frac{m * |A_R|}{\max\left(|A_R|, \left|\widehat{A}\right|\right)} \tag{4}$$

In our evaluation, we use $m = 1$, and calculate the weighted average for each split by using $\frac{|A_R|}{\sum_{i=1}^{N}|A_R^i|}$ as the weight of each instance. Following evaluation rules in TEACh (Padmakumar et al., 2021), the maximum number of action steps is 1,000 and the failure limit is 30.

**Baseline** We select Episodic Transformer (E.T.) (Padmakumar et al., 2021) as the baseline method. E.T. achieved SOTA performance on the TEACh dataset. It learns the action prediction based on the TEACh dataset using a ResNet-50 (He et al., 2016) backbone to encode visual observations, two multi-modal transformer layers to fuse the embedded language, visual and action information, and output the executable action directly.

## Main Results and Analysis

**EDH** We establish the state-of-the-art performance with a large margin over the baseline E.T.model. Our model achieves a higher relative performance than baselines on trajectory-weighted metrics, as shown in Table 1, suggesting that adding symbolism can also reduce navigation trajectory length. Besides, our framework has less performance gap between seen and unseen environments, showing better generalizability in new environments. We find that E.T. usually gets stuck in a corner or keeps repeatedly circling, while our model barely suffers from those issues, benefiting from neuro-symbolic commonsense reasoning.

**TfD** Our model achieves state-of-the-art performance across all metrics in TfD tasks, as shown in Table 1, which shows that JARVIS has better capability for task execution based on offline human-human conversation. Compared with EDH tasks, TfD tasks provide only the entire dialog history for the agent, which increases the difficulty and causes performance decreases in both models, while our JARVIS still outperforms the E.T. by a large margin, showing that the end-to-end model can not learn an effective and generalized strategy for long tasks completion providing only dialog information.

**TATC** We evaluate our framework on the TATC task with a *Commander* under three distinct constraint levels, simulating varied human assistance. The *Commander* is provided

with: 1) only the current sub-goal; 2) the sub-goal and target object location; or 3) the sub-goal, location, and segmentation mask (the original TATC setting Padmakumar et al. (2021)). As shown in Table 2, JARVIS greatly outperforms the only open-source baseline, which fails on all TATC instances, highlighting the task's complexity for end-to-end methods. Furthermore, the Success Rate (SR) improves as the *Commander* is given more information, demonstrating that JARVIS effectively adapts its instructions to leverage all available knowledge.

**Few-Shot Learning**

Data scarcity has been known as a severe issue for deep neural methods. Especially, it is even more severe in language-involved embodied agent tasks since collecting training data is more expensive and time-consuming. Here we also conduct experiments in the few-shot setting, shown in Table 1. We randomly sample ten instances from each of the 12 types of household tasks in the TEACh dataset and train the language understanding and planning module and Goal Transformer in the same way as the whole dataset setting. For E.T., we notice a significant performance drop on both EDH and TfD (e.g., 0 success rate in TfD tasks), since it overfits and can not learn effective and robust strategy. Since our framework breaks down the whole problem into smaller sub-problems and incorporates a solid symbolic commonsense reasoning module, it still has the ability to complete some complex tasks. This also indicates the importance of connecting connectionism and symbolism.

**Unit Test of Individual Module**

To analyze the performance gain of JARVIS, we conduct ablation studies on EDH and TfD tasks (Table 4). With ground truth perception and sub-goals, the Symbolic Commonsense Reasoning module achieves over 60% success in EDH, showing its effectiveness in action inference when provided accurate inputs. Replacing our language planner with ground truth yields greater improvement than replacing the executor, confirming the importance of symbolic reasoning in short-horizon tasks.

In longer TfD tasks, where a single sub-task failure leads to overall task failure, executor quality becomes the bottleneck. Here, replacing our executor with a teleport agent boosts performance more than using ground truth sub-goals. We also observe that our Goal Transformer improves trajectory-weighted success, indicating more efficient action planning.

## 5. Conclusion

This work studies how to teach embodied agents to execute dialog-based household tasks. We propose JARVIS, a neuro-symbolic framework that can incorporate the perceived neural information from multi-modalities and make decisions by symbolic commonsense reasoning. Our framework outperforms baselines by a large margin on three benchmarks of the TEACh (Padmakumar et al., 2021) dataset. We hope the methods and findings in this work shed light on the future development of neuro-symbolic embodied agents.

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

## Appendix A. Implementation Details

JARVIS takes advantage of both connectionism and symbolism. Here we first introduce the learning details of the deep learning modules in JARVIS, followed by the symbolic commonsense reasoning. Then we introduce the task completion process of the JARVIS framework, shown in the Algorithm 1.

### Learning Modules

Here we elaborate on the implementation details of the learning and symbolic reasoning modules in our framework, including language understanding and planning, semantic world representation, and goal transformer.

#### Language Understanding and Planning

For data collection of the EDH task, we collect one data sample from each EDH instance. In each data sample, the input contains a history dialogue between the commander and the follower and history sub-goals that have been executed, and the output is the future sub-goals. We create special tokens $< COM >, < FOL >$ to concatenate different utterances of the dialogue, and $< HIS >$ to concatenate the history sub-goals and dialogue. Note that the history sub-goals are not available in the training set, so we translate the provided history actions into history sub-goals by excluding all the navigation. For the TfD task, we collect the whole dialog sequence and the whole ground truth future sub-goal sequence from each TfD instance as input and output for training. We collect the same data scheme from seen validation EDH/TfD instances for validation. We adapt the pre-trained BART-LARGE Lewis et al. (2020) model and fine-tune the BART model separately on the collected training data for each task. We follow the Huggingface Wolf et al. (2019) implementation for the BART model as well as the tokenizer. The BART model is finetuned for 50 epochs and selected by the best validation performance. We use the Adam optimizer with a learning rate of $5 \times 10^{-5}$. The maximum length of generated subgoal is set to 300, and the beam size in beam search is set to 4.

#### Semantic World Representation

For Mask-RCNN He et al. (2017), we use a model pre-trained on MSCOCO Lin et al. (2014) data and fine-tune it on collected data from training environments of TEACh dataset. We get the ground truth data samples from the provided interface of AI2THOR. During training, the loss function is as follows:

$$L = L_{cls} + L_{box} + L_{mask} \tag{5}$$

Where $L_{cls}$ is the cross entropy loss of object class prediction. $L_{box}$ is a robust $L_1$ loss of bounding box regression. And $L_{mask}$ is the average binary cross entropy loss of mask prediction, where the neural network predict a binary mask for each object class. Following Shridhar et al. (2021), we use the batch size of 4 and learning rate of $5 \times 10^{-3}$ for Mask-RCNN training. We train for 10 epochs and select by the best performance on data in unseen validation environments.

For the depth prediction model, we use an Unet-based model architecture same as Blukis et al. (2021). We get the images and ground truth depth frames from the training environments of TEACh dataset by AI2THOR Kolve et al. (2017) too. The range of depth prediction is from 0-5 meters, with an interval of 5 centimeters. Thus, the depth prediction problem is formulated as a classification problem over 50 classes on each pixel. During training, the loss function is a pixel-wise cross entropy loss:

$$L = \text{CE}(D_{predict}, D_{gt}) \tag{6}$$

$D_{gt}$ stands for ground truth depth and $D_{predict}$ stands for predicted depth. During training, we follow the training details in Blukis et al. (2021) and using the batch size of 4, learning rate of $1 \times 10^{-4}$. We train for 4 epochs and select by the best performance on data in unseen validation environments.

For semantic map construction, we first project the depth and semantic information predicted by learned models into a 3-D point cloud voxel, based on which we do vertical projection and build a semantic map of $240 \times 240$ for each object class and obstacle map. Each pixel represents a $5cm \times 5cm$ patch in the simulator.

## Goal Transformer

We collect one data sample from each EDH instance to train the Goal Transformer. The Goal Transformer takes the ground truth future sub-goals, history images, and history actions as inputs. We modify the prediction head of the Episodic Transformer to generate the actions in TEACh benchmarks Padmakumar et al. (2021). The Goal Transformer uses a language encoder to encode the future sub-goals, an image encoder (a pre-trained ResNet-50) to encode the current and history images, and an action embedding layer to encode the history actions in order. The model is trained to predict future actions autoregressively. During training, we use the cross entropy loss between the ground truth future action $A_{gt}$ and predicted future action, as is $A_{predict}$:

$$L = \text{CE}(A_{predict}, A_{gt}) \tag{7}$$

We follow the episodic transformer (E.T.) Pashevich et al. (2021) training details for the goal transformer model. The batch size is 8 and the training epoch is 20. We use the Adamw optimizer Loshchilov and Hutter (2017) with a learning rate of $1 \times 10^{-4}$.

## Symbolic Reasoning

Here, we provide some details of the implementations of the symbolic reasoning part. In task-level commonsense reasoning, we mainly check sub-goals from two perspectives: properties and causality. For properties, we need to check whether the action is affordable for the object. Therefore, we define some object collections depending on the properties, including movable, sliceable, openable, toggleable, and supportable. We can determine the unreasonable sub-goals by checking whether the planned object can afford the corresponding action. Then, causality means whether the sub-goal sequence obey the causal relations, like "a knife in hand" should always be the prerequisite of "slice a bread". To achieve this purpose, we define some rules of prerequisites and solutions according to commonsense,

including placing the in-hand object before picking something, removing the placing action if nothing in hand, etc. To check the prerequisites, we assume all the previous sub-goals are completed and check if the agent's states matches the desired states. If the current sub-goal and the agent's state have causal conflict, we will use predefined solutions to deal with it. For example, we will remove *Place(agent, x)* if there is nothing in the hand. We will add *Pick(agent, "Knife")* before *Slice(agent, "Bread")* if the agent do not grasp a knife in hand. The whole process can be found in Algorithm 2.

We consider both the semantic map and action states for action-level commonsense reasoning. In general, the agent updates the semantic map according to the observation of every step. We use the pixels change to detect whether the previous action success. To determine the next step, the agent must check whether the target object has been observed. If the target object has been observed, an FMM algorithm will plan a path to the closest feasible space. Otherwise, we will use the Goal Transformer to determine the next action for exploration. The action space of the Goal Transformer is all motion actions, including forwarding, backward, panning left, panning right, turning right, and turning left. During the movement to the target position, if the agent counterfaces unexpected collisions due to the error of the semantic map, the agent will save the current pose and the action from causing the collision. Then, the agent will first consider using the estimated motion action from Goal Transformer. But if the output action of GT is the same as the previous one leading to the collision, it will take a random motion action. After the agent reaches the target position, the agent will rotate and move around the place if the target object cannot be found in the current observation. If all attempts have been made and the object is still unobservable, the agent will consider it as a false detection situation and add the corresponding signal in the semantic map. The whole process can be found in Algorithm 3.

**Task Completion Process of the JARVIS framework**

We elaborate logic predicates for symbolic commonsense reasoning in Table 5. In Algorithm 1, we show the overall process when our JARVIS framework tries to finish an EDH or TfD tasks, which includes two algorithms to implement symbolic commonsense reasoning predicates: Algorithm. 2 describes semantic map building process and language planning process, which including the task-level commonsense reasoning. Algorithm. 3 describes the detailed action generation process with action-level commonsense reasoning.

**Experiment Details for Two Agent Task Completion (TATC)**

We experiment with our JARVIS framework on TATC task in three different settings, where there are different constraints about how much information will be available to the *Commander* for generating instructions. As in Table. 2, in the first setting named full info. setting, the *Commander* has all information about the current subgoal, eg. *pickup knife*, target object location and the ground truth segmentation of the target object in the view. In this setting, same as the setting in Padmakumar et al. (2021), the *Commander* can specifically instructs where the *Follower* should interact to finish the current subgoal. In the second setting, we eliminate the ground truth segmentation of the target object from the information provided to the *Commander*. As a result, the *Commander* could still instruct the *Follower* to arrive near the target object but will not be able to explicitly tell the

Table 5: Logic predicates for symbolic commonsense reasoning.

| Task Common Sense | | Action Common Sense | |
|---|---|---|---|
| $Movable(x)$ | True if $x$ can be moved by the agent. | $IsEmpty(x)$ | True if location $x$ is empty in map. |
| $Slicedable(x)$ | True if $x$ can be sliced into parts. | $Observe(x)$ | True if $x$ is observed. |
| $Openable(x)$ | True if $x$ can be opened or closed. | $Success(x,y)$ | True if action $x$ can be executed at state $y$. |
| $Toggleable(x)$ | True if $x$ can be toggled on or off. | $Near(x,y)$ | True if the distance from the agent to $x$ is smaller than $y$. |
| $IsReceptacle(x)$ | True if $x$ can support other objects. | $Move(x,y)$ | $\exists P \subset V, \forall i \in P, IsEmpty(i)$ where $V$ is the all possible paths from $y$ to $x$. |
| $IsGrasped(agent)$ | True if agent has grasped something. | $Collision(x,y)$ | $Move(x,y) \land \neg Success(x,y)$ |
| $Pick(agent,x)$ | $\neg IsGrasped(agent)$ $\land Moveable(x)$ | $Target(x,y)$ | $Observe(x) \land Move(x,y)$ |
| $Place(agent,x)$ | $IsGrasped(agent) \land IsReceptacle(x)$ | $Interactive(action,x)$ | $Observe(x) \land Near(x,0.5)$ |
| $Slice(agent,x)$ | $Pick(agent,\text{“}Knife\text{”}) \land Sliceable(x)$ | $Ignore(action,x)$ | $\neg Observe(x) \land Near(x,0.5)$ after exploring around x for serval times. |

---

**Algorithm 1** Task Completion Process of the JARVIS framework

**Input:** History observations $V_{history}$, History agent actions $A_{history}$, Utterance $u$

1  $G_{future}, M_{semantic} \leftarrow Algorithm\ 2(V_{history}, A_{history}, u)$;
2  $pointer \leftarrow 0$;
3  $fail\_times \leftarrow 0$;
4  $step \leftarrow 0$;
5  $need\_stop \leftarrow$ False;
6  $a_{prev} \leftarrow A_{history}[-1]$ ;                          // The previous action is the last in history actions
7  $v_{curr}, success\_execute \leftarrow$ Simulator.Step($a_{prev}$);
8  **while** **not** $need\_stop$ **do**
9      $a_{next}, M_{semantic}, pointer \leftarrow Algorithm\ 3(a_{prev}, v_{curr}, M_{semantic}, pointer, G_{future})$;
10     $v_{curr}, success\_execute \leftarrow$ Simulator.Step($a_{next}$);
11     $a_{prev} \leftarrow a_{next}$;
12     $step \leftarrow step + 1$;
13     **if** **not** $success\_execute$ **and** $a_{prev}$ is an interaction action **then**
14         $\lfloor\ fail\_times \leftarrow fail\_times + 1$;
16     **if** $fail\_times \geq 30$ **or** $step \geq 1000$ **or** $a_{next}$ **is** "Stop" **then**
17         $\lfloor\ need\_stop \leftarrow$ True;

---

follower the ground truth location of the target object in the view. In the third setting, we further restrict the target object location (goal location) from the information to the *Commander* and thus the instruction generated from the *Commander* will not include any ground truth information about where the target object is.

## Appendix B. Notation Table

We describe the notation used in this paper in Table. 6.

## Appendix C. Case Study

In the EDH task, the agent can process history dialog and execute sub-goals, and plan future actions to complete the task. With well-designed Symbolic Commonsense Reasoning module, the agent can efficiently navigate to the target location and execute planned actions, as shown in Figure. 3. In the TfD task, the agent is able to understand the dialog and break down the whole task into future sub-goals, and execute it correctly, as in Figure. 4.

---

**Algorithm 2** Semantic Map Building and Language Planning

---

**Input:** History observations $V_{history}$, History agent actions $A_{history}$, Utterance $u$
**Output:** Future Subgoals $G_{future}$, Semantic map $M_{semantic}$

18   $G_{history} \leftarrow \{\}$; $M_{semantic} \leftarrow \{\}$; $pointer \leftarrow 0$ ;          *// Initialize pointer for subgoals*
19   $need\_nav \leftarrow$ False;

20   **Function** TaskCommonSenseProcess($G$):
21      $picked\_object \leftarrow$ None; **for** $i \leftarrow 0$ **to** $G.length - 1$ **do**
22         $a_i \leftarrow G[i]$.action; $object_i \leftarrow G[i]$.target$\_$object; **if** $a_i$ **is** `PickUp` **then**
23             **if** Movable($object_i$) **then**
24                **if** $picked\_object$ **is not** *None* **then**
25                  Add [Place, CounterTop] into $G$ before step $i$;
26                $picked\_object \leftarrow object_i$;
27             **else**
28                Remove $a_i$ from $G$;
29         **else if** $a_i$ **is** `Place` **then**
30             **if** IsReceptacle($object_i$) **then**
31                $picked\_object \leftarrow$ None;
32             **else**
33                Remove $a_i$ from $G$;
34         **else if** $a_i$ **is** `Slice` **then**
35             **if** Sliceable($object_i$) **then**
36                **if** $picked\_object$ **is not** *"Knife"* **then**
37                  **if** $picked\_object$ **is not** *None* **then**
38                    Add [[Place, CounterTop], [PickUp, Knife]] into $G$ before step $i$;
39                  **else**
40                    Add [PickUp, Knife] into $G$ before step $i$;
41                $picked\_object \leftarrow$ "Knife";
42             **else**
43                Remove $a_i$ from $G$;
44         **else if** *($a_i$ **is in** [`Open`, `Close`] **and not** Openable($object_i$)) **or***
            *($a_i$ **is in** [`ToggleOn`, `ToggleOff`] **and not** Toggleable($object_i$))* **then**
45             Remove $a_i$ from $G$;
46      **return** $G$;

47   **for** $i \leftarrow 0$ **to** $t - 1$ **do**
48      $a_i \leftarrow A_{history}[i]$; $v_i \leftarrow V_{history}[i]$; **if** $a_i$ **is in** *interaction$\_$action* **then**
49         $Target\_object \leftarrow a_i$.target$\_$object; **if** $need\_nav$ **then**
50             $G_{history} \leftarrow G_{history} +$ [Navigate, $Target\_object$] $+$ [$a_i$.action$\_$name, $Target\_object$];
51         **else**
52             $G_{history} \leftarrow G_{history} +$ [$a_i$.action$\_$name, $Target\_object$];
53         $need\_nav \leftarrow$ False;
54      **else**
55         $need\_nav \leftarrow$ True; $M_{semantic} \leftarrow$ ObservationProject($v_i, a_i, M_{semantic}$);

56   $G_{future} \leftarrow$ LanguagePlanner($G_{history}, u$);
57   $G_{future} \leftarrow$ TaskCommonSenseProcess($G_{future}$);
58   Initialize Goal$\_$Transformer with $V_{history}, A_{history}, G_{history}$;

---

We also show an example of one classic error of episodic transformer in 3(*b*). The E.T. model will repetitively predict "Forward" when facing the wall, or "Pickup Place" in some other cases. This is because the agent can not correctly and robustly infer correct actions from all the input information.

Figure. 5 illustrates how our JARVIS framework can be adapted to the TATC task under the setting that the commander can acquire state changes needed to be complete,

---

**Algorithm 3** Action Generation with Action Commonsense Reasoning

---

**Input:** Previous action $a_{t-1}$, Current observation $v_t$, Semantic map $M_{t-1}$, Subgoal pointer $pointer_{t-1}$, Future subgoal $G_{future}$

**Output:** Next action $a_t$, Updated semantic map $M_t$, Updated subgoal pointer $pointer_t$

59  $stop\_navigate \leftarrow$ False;
60  $a_t \leftarrow$ None;
61  $previous\_success \leftarrow$ CheckSuccess($v_t$) ;                             // *True if pixel changes ¿ threshold*
62  **if** $previous\_success$ **then**
63     $pointer_t \leftarrow pointer_{t-1} + 1$; ExecutePostActionProcess($a_{t-1}$) ;          // *Update planner states*
64     $M_t \leftarrow$ ObservationProject($v_t, a_{t-1}, M_{t-1}$);
65  **else**
66     $pointer_t \leftarrow pointer_{t-1}$; $M_t \leftarrow$ UpdateCollision($a_{t-1}, M_{t-1}$);

67  $g_t \leftarrow G_{future}[pointer_t]$; $Goal\_ET\_action \leftarrow$ Goal_Transformer.GetNextAction($v_t, a_{t-1}, g_t$); $object_t \leftarrow g_t$.target_object; $target\_observed \leftarrow$ Observe($object_t$) ;         // *True if target is in current $v_t$*
68  $target\_found \leftarrow$ FindInMap($object_t, M_t$) ;                          // *True if target is in $M_t$*
69  **if** $target\_found$ **then**
70     $stop\_navigate \leftarrow$ Near($object_t, 0.5$) ;                   // *True if distance to target ¡ 0.5m*
71     **if** $stop\_navigate$ **then**
72        **if** $target\_observed$ **then**
73           $a_t \leftarrow$ Interaction($g_t$.action, $object_t$);
74        **else**
75           $a_t \leftarrow$ FindTargetNearDestination($M_t$) ;      // *Perform delta moves around destination*
76     **else**
77        $a_t \leftarrow$ FMM($object_t, M_t$);
78  **else**
79     $a_t \leftarrow Goal\_ET\_action$;
80  **if** **not** $previous\_success$ **and** $a_t$ **is equal to** $a_{t-1}$ **then**
81     $a_t \leftarrow$ GenerateRandomFesibleAction($M_t$) ;                 // *Random non-colliding movement*
82  **return** $a_t, M_t, pointer_t$

---

Table 6: Primary notation table.

| Symbol | Description |
|--------|-------------|
| $\mathcal{A}$ | The action space. |
| $A$ | A random variable representing the action sequence. |
| $a_t$ | A specific action at time $t$. |
| $\int$ | The environment state space. |
| $s_i$ | The initial state. |
| $s_f$ | The final state. |
| $D$ | Dialogues consisting of pairs of players and utterance. |
| $p$ | The players which can only be *Commander* or *Follower*. |
| $u$ | The utterance. |
| $V$ | A random variable representing the visual input of the *Follower*. |
| $G$ | The sequence of sub-goals which is an intermediate guidance used for making decisions. |

ground truth segmentation, and object location. The symbolic commonsense reasoning action module of our JARVIS-based follower can generate actions to complete the sub-goals provided by the commander.

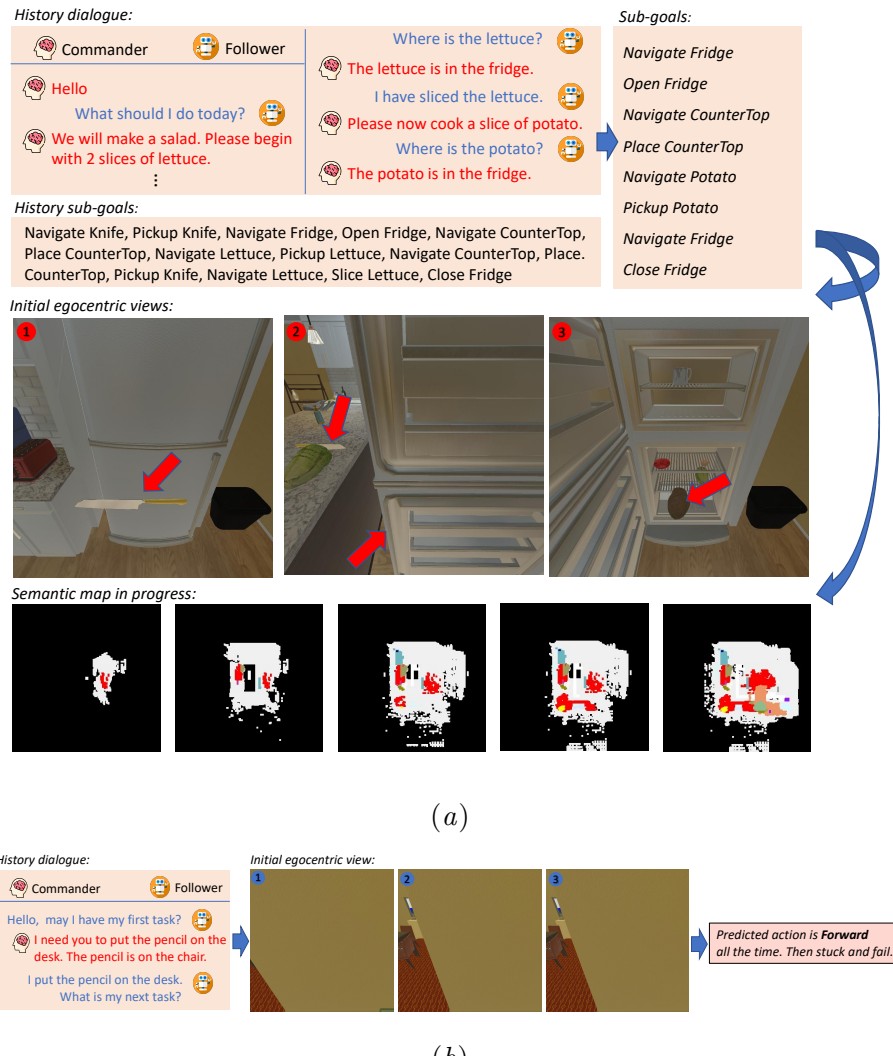

(a)

(b)

Figure 3: EDH example. (a) shows an example of our JARVIS in EDH task, where the inputs are dialog history and sub-goal history (converted from action history input). The inputs are first interpreted by the Language Parsing Module to become sub-goals. Then, our Symbolic Reasoning Module will generate action predictions. The predicted actions will change the follower's egocentric views and the semantic map will be built up and completed gradually. ① shows the agent is opening the fridge ② shows the agent has placed the knife and navigate back to the fridge. ③ shows the agent is picking up the potato. (b) is an example demonstrating a typical way of how Episodic Transformer fails on EDH task. In this case, the E.T. model predicts "Forward" repetitively even facing the wall, therefore stuck at the current position.

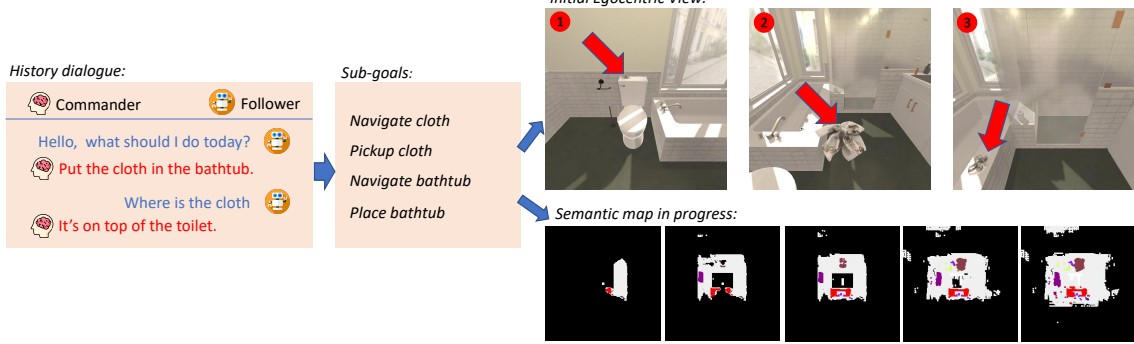

Figure 4: Successful TfD example from our JARVIS framework. According to the dialog, the language planner estimate four future sub-goals: ("Navigate cloth", "PickUp Cloth", "Navigate Bathtub", "Place Bathtub"). Then with the symbolic reasoning module, interaction and navigation actions are predicted. ❶ shows the agent is finding the cloth. ❷ shows the agent has picked up the cloth and then found the bathtub. ❸ shows the agent can correctly put the cloth on the bathtub.

Table 7: Average action failure rate on validation set for EDH, TfD and TATC tasks. When the time of action failure in a session reaches 30, the session will be forced to end, causing a task failure. The action failure exist widely in sessions.

| Failure Mode | EDH(%) | TfD(%) | TATC(%) |
|---|---|---|---|
| No action failure | 4.0 | 17.0 | 5.8 |
| Action failures exist but are less than 30 times | 78.8 | 79.9 | 69.0 |
| Action failures reaches 30 times | 17.2 | 3.1 | 25.2 |

## Appendix D. Error Analysis

According to Table. 7, we find that all failed tasks include at least one action failure. For the shorter tasks, like EDH, the dominant situation is "Action failures are less than 30 times", while the longer tasks, like TfD and TATC, include even more action failures due to more sub-goals. To this aspect, we further analyze the action failure reason as in Table. 8, which shows a quantitative analysis of the failed actions categories in the validation set. We randomly sampled 50 failure episodes in both seen and unseen splits and computed the average ratio of each action failure category in all the failed actions. From Table. 8, we notice that 'Blocked when moving' is the most frequent cause of failure action in all the three tasks, EDH, TfD, and TATC. The main reason is that errors in depth prediction cause semantic obstacle map errors. Besides, 'Too far to interact' is also mainly caused by wrongly predicting the 3D location of an object. To solve this, we need to improve the precision of the depth prediction model or design a more delicate and robust algorithm to update the semantic map. The second frequent failure action is 'Object not found at the location or cannot be picked up,' which could be due to the wrong predicted location of an object by Mask R-CNN. Moreover, it is also likely caused by the situation where the

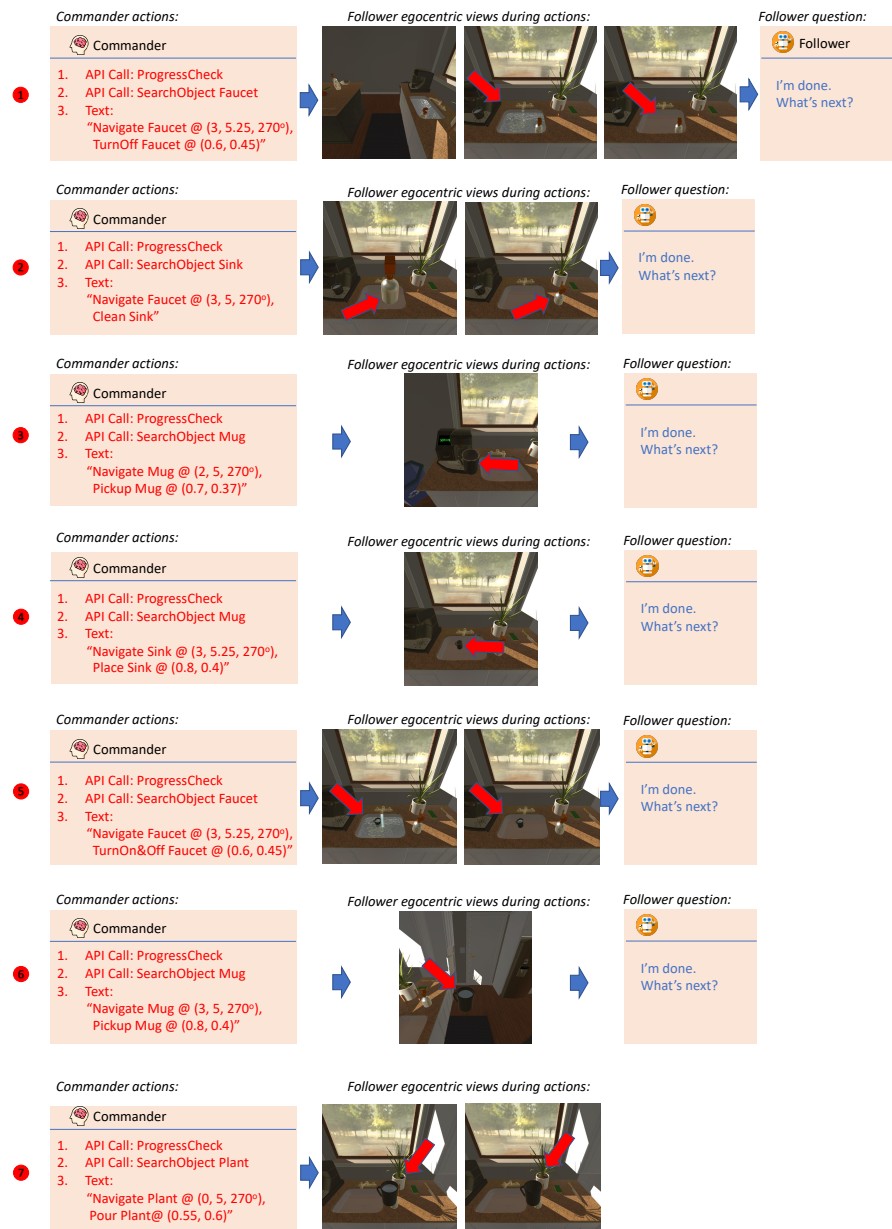

Figure 5: TATC "Water the plant" sequence. At each step the commander calls SearchObject to get an optimal navigation pose (e.g. ❶: Navigate to Faucet @ (3, 5.25, 270°)), then uses ground-truth segmentation to compute an interaction target (e.g. ❸: Pickup Mug @ (0.7, 0.37)). Steps 1–6 repeat for Faucet, Sink, Mug, Sink, Faucet, and Mug; step 7 pours water on the Plant @ (0.55, 0.6). After each sub-goal the follower executes in its egocentric view and reports completion.

target object is inside a receptacle that needs to be open first, as in Figure. 6(a). 'Invalid position for placing the held object' is also mainly caused by the segmentation error of Mask

Table 8: Action failure categorization result on a sub-set of validation set for all three tasks. The numbers show the rate of a certain failure action occurs among the total action failure time.

| Action failure reason | EDH(%) | TfD(%) | TATC(%) |
|---|---|---|---|
| Hand occupied when picking up | 3.20 | 1.46 | 1.04 |
| Knife not in hand when cutting | 3.12 | 1.82 | 1.36 |
| Target object not support open/close | 0.21 | 0.36 | 0.83 |
| Cannot open when running | 0.07 | 0.80 | 0.31 |
| Blocked when moving | 40.81 | 40.74 | 47.34 |
| Collide when rotating | 5.68 | 3.57 | 0.63 |
| Collide when picking up | 0.14 | 0.00 | 0.10 |
| Invalid position for placing the held object | 9.37 | 7.94 | 4.28 |
| Too far to interact | 8.52 | 10.13 | 4.28 |
| Pouring action not available for the held object | 1.13 | 1.82 | 0.10 |
| Object not found at location or cannot be picked up | 21.50 | 19.39 | 28.26 |
| Cannot place without holding any object | 6.17 | 11.30 | 8.03 |
| Collide when placing | 0.00 | 0.00 | 0.21 |
| Target receptacle full when placing | 0.07 | 0.66 | 3.23 |

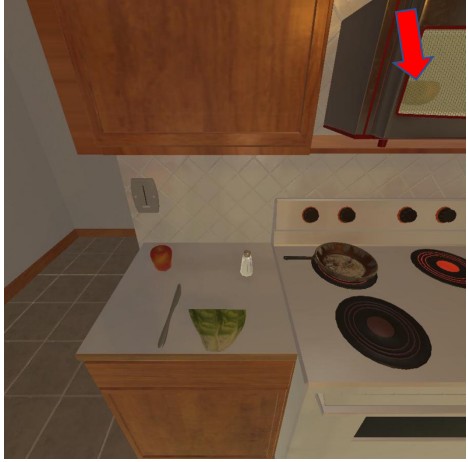
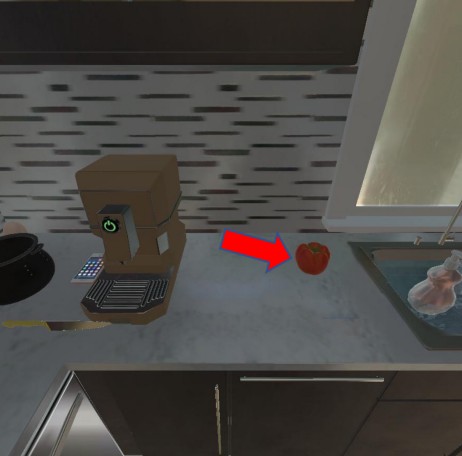

(a) Object not found at location or cannot be picked up  (b) Miss-recognized target object

Figure 6: Action failure examples

R-CNN. Thus, improving the ability of two perception models is the most effective way to avoid failure actions.

Other frequent reasons for failure are 'Cannot place without holding any object,' 'Hand occupied when picking up,' and 'Knife not in hand when cutting.' These are because of the wrong judgment of whether the former 'Place' and 'Pickup' actions have succeeded. For example, the agent might think it has picked up the knife, while it failed to pick up in fact,

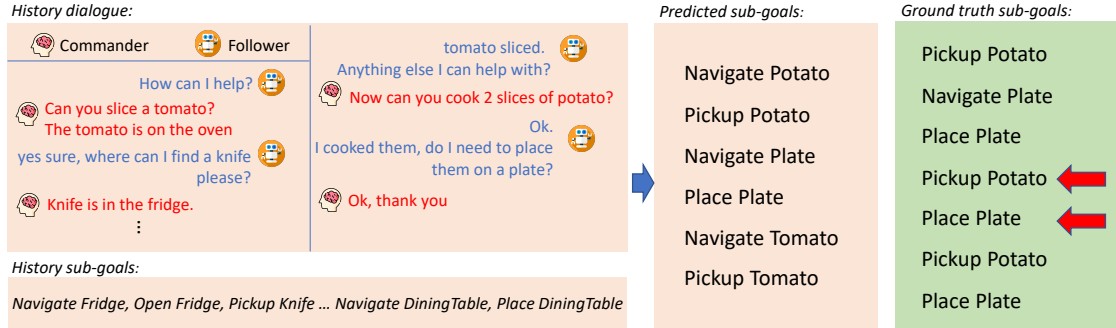

Figure 7: An example of sub-goal estimation failure. The predicted sub-goal lacks two important sub-goal as pointed by the red arrows, which causes the task failure in the end.

which causes the latter 'Knife not in hand when cutting.' We need to further refine our Symbolic Commonsense Reasoning Action module for these cases.

Instead of the failed actions, a miss-recognized object will also lead to an unsuccessful task. As in Figure 6(b), the tomato is falsely recognized as an apple, which leads to a failure when the target object is about "Tomato". Additionally, the false sub-goals estimations also contribute to the failures in task completion. In Section 4, we quantitatively analyze the performance of the Language Planning Module in the JARVIS framework. Here, we show a typical qualitative error result in Fig. 7. According to the dialog, the ground truth future sub-goals need the follower to put all potatoes on the plate. However, the estimated sub-goals indicate one slice of potatoes and tomatoes will be put on the plate, which will cause the unsatisfied state changes.

