# OpenReview forum: "JARVIS: A Neuro-Symbolic Commonsense Reasoning Framework for Conversational Embodied Agents"
_nesyconf.org/NeSy/2025/Conference_Phase_2 — NeSy 2025 - Phase 2 Poster_

### Official Review · Reviewer_2iVH · 2025-06-28
**Promising Neuro-Symbolic Framework with Strong Results, but Limited Experimental Scope**

**Rating:** 6
**Confidence:** 4

**Review:**

This paper proposes JARVIS, a modular neuro-symbolic framework for conversational embodied agents that combines large language models, visual semantic mapping, and symbolic commonsense reasoning. Evaluated on the TEACh benchmark, it achieves state-of-the-art performance across multiple tasks and demonstrates strong generalization in few-shot settings.

In the proposed method, the modular design allows flexible adaptation to different dialog-based task settings, where the integration of LLMs for sub-goal generation from free-form dialogue is pragmatic and shows strong empirical utility, and the symbolic reasoning module enhances both interpretability and execution robustness, especially in scenarios where the language or perception modules introduce noise. The paper provides extensive experimental validation, including few-shot settings, ablation studies, and detailed error analysis, demonstrating both the effectiveness and generalizability of the framework.

A key limitation is that the evaluation is confined to the TEACh benchmark, which focuses solely on simulated household tasks. This raises concerns about the framework's applicability to other domains. Especially, the symbolic reasoning module depends on a handcrafted rule set that is specifically tailored to TEACh, and may not generalize well to tasks involving different object types, action spaces, or environmental dynamics.

**Anonymity:**

Disclose identity

---

### Official Review · Reviewer_RAWT · 2025-07-08
**Strong submission**

**Rating:** 7
**Confidence:** 3

**Review:**

This paper proposes JARVIS - a framework for conversational embodied agents that incorporates three components: planning module, semantic mapping module, and symbolic reasoning for action execution module. The framework is instantiated with small LMs, and it has a positive impact on three tasks: execution from dialog history, trajectory from dialog, and two-agent task completion.

Overall, the paper fits the NeSy theme, and it has sufficient novelty compared to prior work. It is well-structured and mostly easy to follow,  and the results convincingly show the effectiveness of the method across tasks. The ablations with ground-truth information provide further insight into what components are responsible for most mistakes.

I don't see any significant weaknesses with this work. The main thing that would strengthen the paper is some error analysis into what goes wrong, i.e., why is the task performance still fairly low after all the modules are incorporated? Do we need better versions of these modules, or are we missing some functionalities altogether? Table 4 provides some insight into this, but I find it difficult to infer conclusions.
A second improvement on the current paper, related to the first, would be to add a couple of paragraphs with a discussion. Here, the authors can summarize the findings and discuss the limitations of this work.

Minor:
- Section 1 can benefit from citations that support some of the unsupported claims in paragraphs 1 and 2.
- The tables are heterogeneous in size, which hurts the aesthetics of the paper. And they are placed far from the place in the text where they are discussed.

**Anonymity:**

Remain anonymous

---

### Official Review · Reviewer_VXcf · 2025-07-09
**A reasonable approach to commonsense action planning in embodied AI**

**Rating:** 6
**Confidence:** 3

**Review:**

The paper presents a framework to perform commonsense reasoning in the context of an embodied agent that is given everyday tasks. The framework consists of 3 large components: an LLM that translates user instructions and the previous interactions into a set of high-level tasks; a visual model that translates images of the environment (from the agent's perspective) into "semantic maps" that encode where different objects are located; and the core symbolic reasoning module that takes high level goals and the information about the environment and translates it into sequences of executable actions.

The framework seems to be quite robust: while the initial high level goals maybe infeasible, out-of-order, underspecified etc., the reasoning module is able to adapt to these factors.
The module "leverages task-level and action-level commonsense logic to validate and refine execution plans".
Task-level reasoning submodule enforces commonsense axioms onto the high level goals, which are edited if necessary.  --> not enough information about it though
Then the framework generates low-level actions to be performed. The method to generate an action set depends on whether the objects involved are currently visible to the agent, i.e., they are present in the semantic maps. Finally, the action-level reasoning submodule enforces environmental constraints onto the action sequence, which may also be edited.

The overall architecture does make sense, and the best of both worlds (symbolic and subsymbolic) are utilized in an intuitive way.
However, very few details and examples are given about the core reasoning module, although there are more details in the appendix. I would encourage the authors to considerably expand Section 3 -> Proposed Methods -> Action Execution...

Evaluation is done over 3 different tasks; unfortunately, the comparison is only done with one other system, although that system seem to represent the current SOTA. I appreciate that the baseline system is the only one that is openly available for the TEACh dataset, but then perhaps there could be other ways to compare JARVIS with other embodied AI systems out there.

These are my two main criticisms, but I believe the paper is a good fit, it represents an important and realistic problem. I enjoyed the in-depth examples in the Appendix.

Minor:
- page 6: "The visual input into a embodied agent" --> an
- page 6: ,vT}. --> overflowing line
Figure 2 is a bit confusing, especially the placement and the numbering of step 0.2.

**Anonymity:**

Remain anonymous

---

### Official Review · Reviewer_aZw6 · 2025-07-11
**Review - JARVIS**

**Rating:** 7
**Confidence:** 3

**Review:**

### Strengths:
- I very much like how the framework is constructed to allow for consistency checking across multiple different modalities, but there is not one modality in particular that has disproportionate influence on the actions or goal structure.
- I liked how the paper examined three different settings rather than just one, all of which I think are relevant to different important applications. It shows that the framework is robust and capable of being successful in environments with varying levels of available information.

### Weaknesses:
- In section A of the appendix, it appears that much of the symbolic reasoning component is hand-defined. This includes constraints, the object collection, as well as associated properties of said objects. These appear to remain static until they are redefined for a different task. When it comes to adaptability of the system, I think it's crucial that there be some update procedure for the reasoning system, as otherwise I feel the follower would have to rely quite a bit on generalized fallback policies within unfamiliar environments.
- It is not clear to me what the benefit of using a UNet-based depth estimator is. There are several robust and explicit 3D reconstruction algorithms that exist, many of which are more computationally efficient than neural inference. Examples of alternatives include structure from motion, as well as depth from perspective using projective geometry. To utilize these, the full coordinate system of the environment in question doesn't generally need to be known. Rather, computing depth using relative coordinates from various points in the environment tends to be sufficient.

### Overall Recommendation:
- My recommendation is to accept the submission. The paper does a good job describing the end-to-end pipeline of the system, demonstrates SOTA benchmarks, and provides a novel general framework that is effective in multiple different settings of known information. The primary revision I would like to see is for additional discussion to be included regarding subsystem design choices and their associated trade-offs with alternatives (for example, why was UNet chosen, as mentioned above).

**Anonymity:**

Remain anonymous